# Biodiesel Production Using Bauxite in Low-Cost Solid Base Catalyst Precursors

**Yong-Ming Dai [1], Cheng-Hsuan Hsieh [2], Jia-Hao Lin[3], Fu-Hsuan Chen [4] and Chiing-Chang Chen [3,\*]**

[1] Department of Chemical and Materials Engineering, National Chin-Yi University of Technology, Taichung 411, Taiwan; forest1105@gmail.com

[2] Department of Materials Science and Engineering, National Tsing-Hua University, Hsinchu 300, Taiwan; frank8771919@gmail.com

[3] Department of Science Education and Application, National Taichung University of Education, Taichung 403, Taiwan; henry811013@yahoo.com.tw

[4] Department of Political Science, National Taiwan University, Taipei 106, Taiwan; fhchen@mail.ntcu.edu.tw

\* Correspondence: cchen@mail.ntcu.edu.tw; Tel.: +886-4-2218-3406; Fax: +886-4-2218-3560

**Abstract:** Investigation was conducted on bauxite mixed with $Li_2CO_3$ as alkali metal catalysts for biodiesel production. Bauxite contains a high percentage of Si and Al compounds among products. Because of the high expense of commercial materials ($SiO_2$, $Al_2O_3$) that makes them not economical, the method was very recently improved by replacing commercial materials with Si and Al from bauxite. This is one of the easiest methods for preparing heterogeneous transesterification catalysts, through one-pot blending, grinding bauxite with $Li_2CO_3$, and heating at 800 °C for 4 h. The prepared solid-base alkali metal catalyst was characterized in terms of its physical and chemical properties using X-ray powder diffraction and field-emission scanning electron microscopy (FE-SEM). The optimal conditions for the transesterification procedure are to mix methanol oil by molar ratio 9:1, under 65 °C, with catalyst amount 3 wt.%. The procedure is suitable for transesterifying oil to fatty acid methyl ester in the 96% range.

**Keywords:** bauxite; $Li_2CO_3$; transesterification; soybean oil

## 1. Introduction

The need for renewable energy sources to meet ever greater energy requirements is becoming increasingly urgent [1]. Many researchers therefore advocate substituting traditional fuels with renewable alternatives that produce lower emissions of particulate matter and reduce greenhouse effects [2,3]. The current process used for producing fatty acid methyl esters (FAMEs) is the transesterification of vegetable oil catalyzed by alkaline catalysts. In addition, this reaction is associated with some difficulties. First, the catalysts cannot be recovered or reused, therefore they must be neutralized. Second, catalysts should be neutralized and separated from the methyl ester phase after the transesterification reaction, with much wastewater resulting [4,5]. Heterogeneous solid catalysts can be used for overcoming these problems. Heterogeneous solid alkaline catalysts are beneficial because of the good separation of the reaction mixtures and their recyclability. Their low price also decreases the overall production costs [6].

By the combination of silicon and lithium carbonate, our previous research discovered that the lithium silicate compound presented high basicity strength, high chemical stability, and thermo-stability. Such characteristics show extremely high applicability in the catalysis field, especially in transesterification [7]. The $LiAlO_2$ structure has recently been published for transestrification. Under high temperatures, lithium aluminate acquired by calcinating $Li_2CO_3$ in the waste with aluminum

forms high basic solid catalyst [8]. Interestingly, both $Li_4SiO_4$ and $LiAlO_2$ compounds demonstrate that these two metal oxides are the best solid basic catalyst for transestrification with high stability, lower cost, and solid basicity.

Bauxite contains a large percentage of Si and Al compounds. The expense of pure Si and Al makes the materials uneconomical. The method has been improved by replacing commercial materials with Si and Al from bauxite [9]. In recent years, newer and simpler methods based on the of bauxite containing a large percentage of Si and Al compounds have been introduced [8,9]. One simple method is based on a solid-state method at high temperatures, blending other base metals with bauxite, which contains Si and Al. The concept involves producing new solid basic catalysts, with the goal of reusing these new precursors and offering broad applications in various fields [10–12]. When bauxite is used as a low-cost solid base catalyst, it provides a twofold advantage in relation to environmental pollution. The use of low-cost materials in the manufacturing process is valuable for industrial applications, particularly for producing FAMEs through transesterification of various vegetable oils, where massive quantities of solid-alkali catalyst are processed [13–16]. The cheaper catalyst precursor from bauxite has some advantages, which are mainly economic and environmental. Another study we undertook applied solid base catalysts to transesterification. This indicated that solid-alkali catalysts provided high transesterification efficiency [6,15,16].

In this paper, bauxite is used as a precursor to prepare alkali catalysts. The resulting catalyst was experimented in the transesterification reaction. The influence of various experimental methods—such as the catalyst amount, methanol–vegetable oil proportions, and catalyst reusability—on efficiency was evaluated to detect the optimum conditions for the study of biodiesel production.

## 2. Results and Discussion

### 2.1. Characterization of As-Prepared Catalyst

Figures 1 and 2 present the X-ray powder diffraction (XRD) results of catalysts. The structure of the bauxite underwent a phase change to $Li_4SiO_4$ (JCPDS-742145) and $LiAlO_2$ (JCPDS-0440224) according to the solid-state preparation with $Li_2CO_3$. Although bauxite was a complex material and the main peaks of its XRD pattern usually overlaid one another, the major phase change in $Li_4SiO_4$ and $LiAlO_2$ was observed after its calcination. Figure 1 shows different calcination temperatures. At a calcination temperature of 600 °C, $Li_2CO_3$ (JCPDS-870728) structures had strong intensity. The catalysts can be observed to exhibit the diffraction peak characteristics of $Li_4SiO_4$, $LiAlO_2$, and $Li_2CO_3$. For calcination at 800 °C, a stronger intensity was present in the $Li_4SiO_4$ and $LiAlO_2$ phase [17,18].

As result of the above, the XRD patterns of the samples calcined at 600 °C corresponded to the $Li_2CO_3$ phase. When the calcination temperature reached the melting point of $Li_2CO_3$ (650 °C), $Li_2CO_3$ entered a molten state [19]. Subsequently, the catalyst (calcined at 800 °C) exhibits the different XRD diffraction peak due to phase transformation to crystalline $Li_4SiO_4$ and $LiAlO_2$. It was found that the XRD patterns of catalyst (calcined at 800 °C to 1000 °C) were similar. These diffraction peaks indicated the presence of crystalline $Li_4SiO_4$ and $LiAlO_2$. The XRD result is attributable to the phenomenon whereby $Li_4SiO_4$ and $LiAlO_2$ started to form agglomerated blocks in the $Li_4SiO_4$ and $LiAlO_2$ phase at high calcination temperatures. Therefore, regardless of the calcination temperature, the $LiAlO_2$ and $Li_4SiO_4$ phases were achieved.

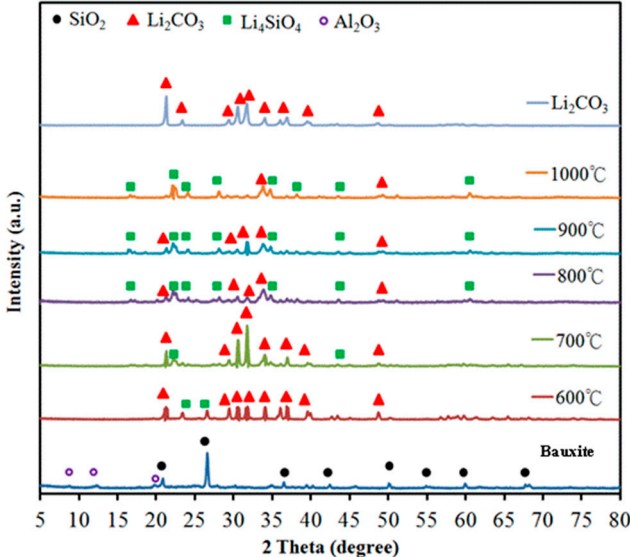

**Figure 1.** X-ray powder diffraction (XRD) of bauxite mixed with $Li_2CO_3$ at different calcination temperatures.

Ordinary bauxite contains two major compounds: $Al_2O_3$ and $SiO_2$. The XRD patterns of prepared catalysts for different bauxite and $Li_2CO_3$ ratios are shown in Figure 2. When the bauxite–$Li_2CO_3$ molar ratio was 0.5, the diffraction peaks of $SiO_2$ were observed. With an increase in the bauxite–$Li_2CO_3$ ratio, the diffraction peaks of $Li_4SiO_4$ and $LiALO_2$ became increasingly clear. When the bauxite–$Li_2CO_3$ ratio increased to 2, the heights of the diffraction peaks belonging to $Li_2CO_3$ increased further. Table 1 presents the efficiency of transesterification to determine the efficiency of all catalysts in the experiment. The reaction test did not provide optimal conditions for the transesterification procedure but provided a method to compare catalytic performance. Results from Table 1 demonstrate that the bauxite exhibited no catalytic performance. When $Li_2CO_3$ was modified bauxite, the catalysts exhibited catalytic activities. The earlier research of solid base catalysts for transesterification reactions is shown in Table 1. Comparing $Li_2CO_3$/Bauxite with other solid base catalysts, it could be clearly found that $Li_2CO_3$/Bauxite showed good catalytic performance for a transesterification reaction.

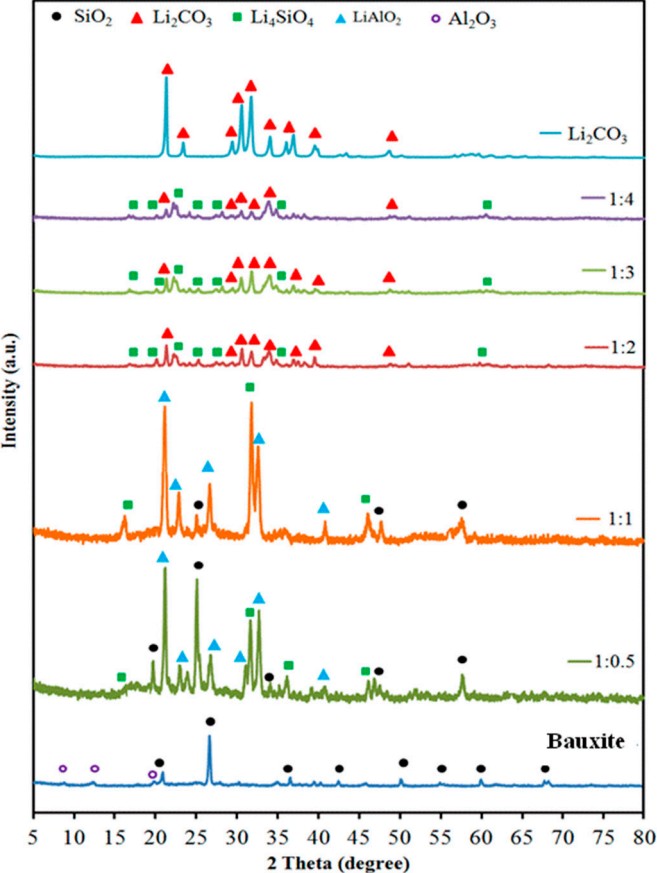

**Figure 2.** XRD patterns of prepared catalyst under different bauxite and $Li_2CO_3$ molar ratios.

**Table 1.** Base strength of bauxite and the prepared catalysts.

| Catalyst | Molar Ratio | Basic Strength | * Conversion (%) |
|---|---|---|---|
| Bauxite | 0 | $7.2 < H^- < 9.8$ | 0.98 |
| $Li_2CO_3$/Bauxite | 1/1 | $9.8 < H^- < 15.0$ | 95.8 |
| $Li_2CO_3$/Bauxite | 1/2 | $9.8 < H^- < 15.0$ | 95.2 |
| $Li_2CO_3$/Bauxite | 1/3 | $15.0 < H^- < 18.4$ | 96.6 |
| $Li_2CO_3$/Bauxite | 1/4 | $15.0 < H^- < 18.4$ | 96.5 |
| $Li_2CO_3$ | - | $9.8 < H^- < 15.0$ | 94.2 |
| $LiAlO_2$ | - | $9.8 < H^- < 15.0$ | 96.4 |
| $Li_4SiO_4$ | - | $9.8 < H^- < 15.0$ | 96.1 |

* Reaction conditions: 12.5 g soybean oil; methanol/oil molar ratio, 12:1; catalyst amount, 6 wt.%; reaction time, 3 h; methanol reflux temperature and conventional heating method.

The structure morphology of catalysts was examined using field-emission scanning electron microscopy (FE-SEM). Figure 3 illustrates morphology noted under FE-SEM for a bauxite-$Li_2CO_3$ molar ratio of 4 calcined in air with various calcination temperatures. The catalyst has a uniform microscale block and agglomerated block composition with a 20–50 μm size range, as shown in Figure 3. Small mineral aggregates and agglomerated circular particles were present when the calcination was performed at high temperatures because of the oxide formation. Figure 3 demonstrates the catalyst results, and circular blocks of different sizes with diameters of approximately 50 μm accumulated on the surface. The particles and amorphous silica disappeared.

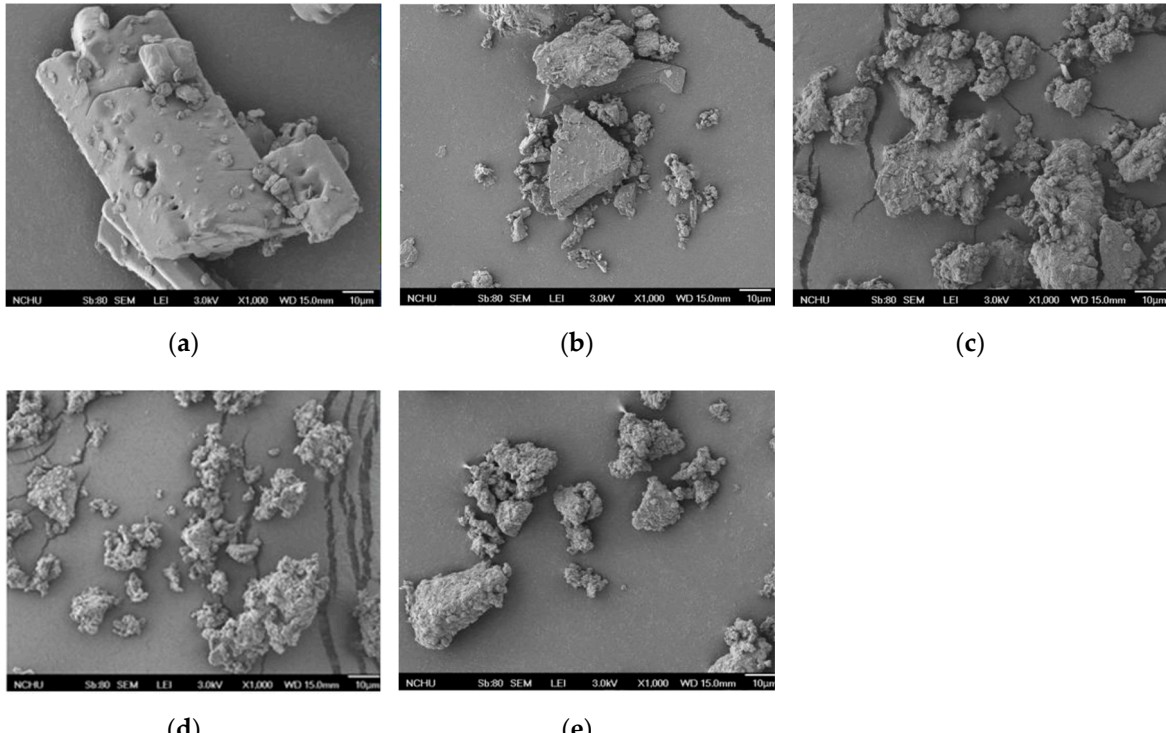

(**a**)         (**b**)         (**c**)

(**d**)         (**e**)

**Figure 3.** Field-emission scanning electron microscopy (FE-SEM) images of bauxite mixed with $Li_2CO_3$ at different calcination temperatures (**a**) 600 °C, (**b**) 700 °C, (c) 800 °C, (**d**) 900 °C, and (**e**) 1000 °C.

## 2.2. Reaction Studies

Experimental tests on how catalyst amount, reaction time, methanol–vegetable oil, and reusability of the catalysts affect efficiency have been evaluated to detect the optimum conditions for the study of biodiesel production.

Figure 4 shows different calcination temperatures and times, for a bauxite-$Li_2CO_3$ molar ratio of 4, through the transesterification process. The reaction rate is shown in the results. With the calcination temperature between 600 °C and 1000 °C, the reaction rate of soybean oil increases to the maximum value, and the reaction rate then begins to decrease when the calcination temperature exceeds 800 °C, as shown in Figure 4. Conversion is consequently lower because catalysts begin to form agglomerated blocks at higher calcination temperature. Hence, the optimal condition occurs at 800 °C in this study. Furthermore, the conversion is found to increase with the increase in time from 1 to 5 h and then to decrease as time elapses past 3 h. In Figure 4, the reaction efficiency suggests that FAME production efficiency is lower at lower calcination times and increases when the calcination time reaches the maximum value of 3 h. The conversion rate then decreases, probably due to the formation saponification for longer calcination times. This result is attributed to the fact that $Li_4SiO_4$ and $LiAlO_2$ start to form agglomerated blocks in the $Li_4SiO_4$ and $LiAlO_2$ phases for long calcination times. Therefore, regardless of the calcination time, the $LiAlO_2$ and $Li_4SiO_4$ phase is achieved. A correct calcination time was consequently required to guarantee completion.

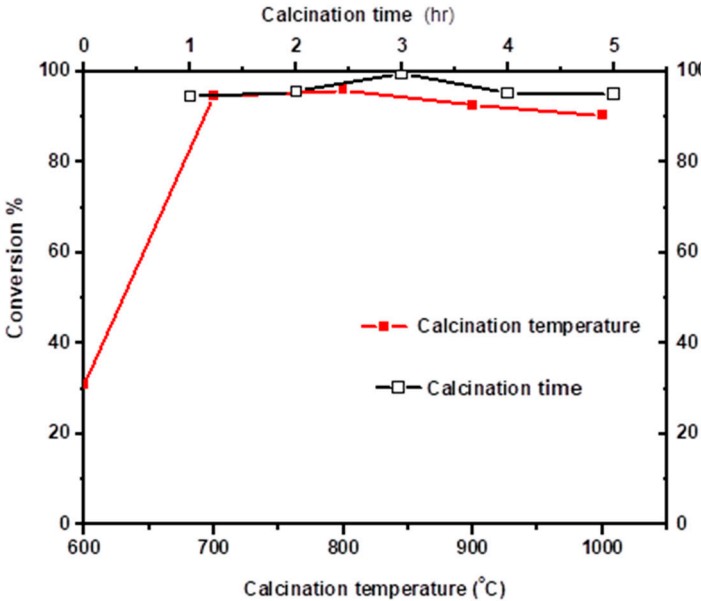

**Figure 4.** Influence of calcination temperature (°C) and time on the conversion reaction conditions: methanol/oil molar ratio 12:1, with 6 wt.% catalyst loading, reaction time 3 h.

Table 2 presents the efficiency for a bauxite and $Li_2CO_3$ molar ratio of 4. As shown, the catalyst exhibits a much higher conversion rate than the latter. In addition, increasing the molar ratio of oil to methanol increased the reaction rate (Table 2). Methanol is a reactant for transesterification reactions [8]. Increasing the reactant quantities shifts the equilibrium to the products side [20]. A molar ratio of 6 yielded a lower conversion rate. The highest reaction rate was obtained at a methanol–oil molar ratio of 12. This molar ratio of 12 was considered the optimal condition and was favorable for the transesterification procedure. Throughout the procedure, the catalyst amount represented a crucial parameter for high efficiency. The catalysts possessing strong main activity sites and a high surface area should exhibit higher conversion. Catalytic sites for transesterification are too few when the catalyst–oil ratio is too low. Catalyst amount was varied from 2 to 10 wt.% to oil to evaluate its effect on the conversion rate of the transesterification procedure at 65 °C and a methanol–oil molar ratio of 12. Catalysts loading of 2 wt.% of oil yielded a lower transfer rate. The maximum efficiency was obtained at a catalyst loading of 6 wt.% (Table 2). A catalyst loading of 6 wt.% for oil was considered the optimal condition and favorable for the transesterification process. This may have been due to the formation of resistance to mass transfer [21] for high catalyst quantities. In addition, to determine the catalytic activity of catalysts in the experimental tests, the efficiency of transesterification is presented in Table 2. This indicates that the reaction rate increased with increasing reaction time in the alkali silicate catalysts at a constant reaction temperature. The transesterification comprised three processes: The first process, whereby triglyceride reacted with one molecule of methanol, yielded diglyceride and one molecule of ester. The second process, the reaction of the diglyceride with a second molecule of methanol, yielded monoglyceride and an additional molecule of ester. In the third process, monoglyceride reacted with the third molecule of methanol, yielding glycerin and ester. Consequently, correct transesterification was required to guarantee completion of the reaction. Table 2 demonstrates that the efficiency increases with time and has an optimum value after 3 h.

**Table 2.** The catalytic performance of catalysts for transesterification of soybean oil with methanol.

| Mole Ratio of Methanol/Oil | Catalyst Amount (wt.%) | Reaction Time (hr) | Oils | * Conversion (%) |
|---|---|---|---|---|
| 6 | 6 | 3 | Soybean oil | 86.3 |
| 12 | 6 | 3 | Soybean oil | 95.9 |
| 18 | 6 | 3 | Soybean oil | 94.1 |
| 24 | 6 | 3 | Soybean oil | 92.1 |
| 12 | 2 | 3 | Soybean oil | 92.4 |
| 12 | 4 | 3 | Soybean oil | 95.2 |
| 12 | 6 | 3 | Soybean oil | 95.9 |
| 12 | 8 | 3 | Soybean oil | 95.0 |
| 12 | 10 | 3 | Soybean oil | 94.1 |
| 12 | 6 | 1 | Soybean oil | 95.6 |
| 12 | 6 | 2 | Soybean oil | 95.8 |
| 12 | 6 | 3 | Soybean oil | 95.9 |
| 12 | 6 | 4 | Soybean oil | 93.1 |
| 12 | 6 | 5 | Soybean oil | 91.1 |
| 12 | 6 | 3 | Coconut oil | 89.1 |
| 12 | 6 | 3 | Olive oil | 95.2 |
| 12 | 6 | 3 | Castor oil | 63.1 |
| 12 | 6 | 3 | Rapeseed oil | 95.4 |
| 12 | 6 | 3 | Corn oil | 95.1 |
| 12 | 6 | 3 | Waste Cooking oil | 92.2 |

*Reaction conditions: 12.5 g oils; reaction temperature, 65°C; methanol reflux temperature and conventional heating method.

The efficiencies of oils other than FAME are indicated in Table 2. FAME was successfully synthesized from soybean oil through a transesterification procedure with catalysts using a simple method. The results demonstrate that the efficiency with high free fatty acid concentration was significantly influenced [22].

The reusability for a bauxite–Li2CO3 molar ratio of 4 was evaluated for soybean oil. The catalyst was reusable up to the sixth repetition, with retention of catalyst efficiency, providing a conversion efficiency of >90% and then declining to a conversion efficiency of 80% after the 6th run repetition (Figure 5).

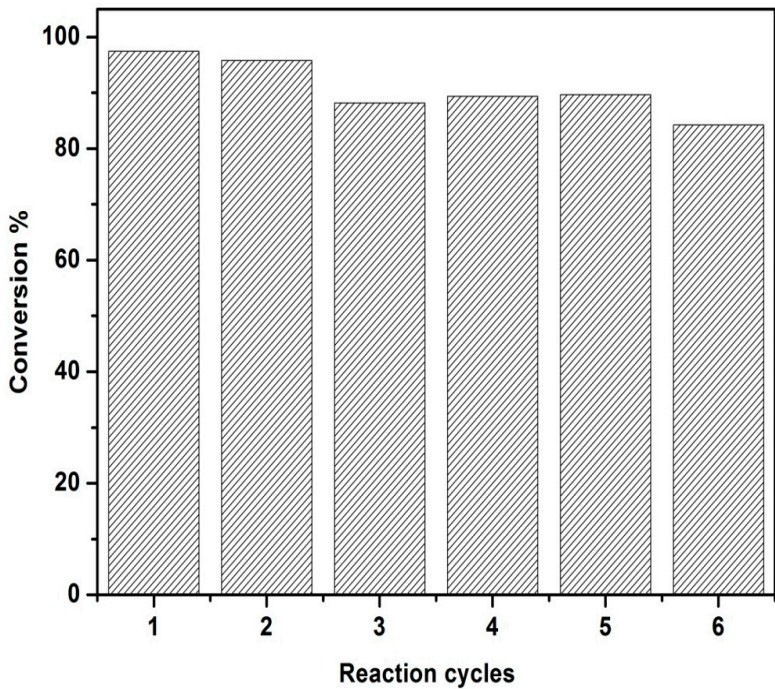

**Figure 5.** Reusability study after six reaction cycles for catalyst. Reaction conditions: methanol/oil molar ratio 12:1, with 6 wt.% catalyst loading, reaction time 3 h.

## 3. Experimental Methods and Chemical Materials

### 3.1. Solid-Alkali Catalyst Preparation

A sample of bauxite was converted into bauxite ash by heat-treating them at 500 °C for 4 h in a muffle furnace. After washing the bauxite ash using deionized water and filtration, the bauxite ash was dried at 120 °C for 16 h. The catalyst was prepared by grinding bauxite ash and $Li_2CO_3$ (Shimakyu's Pure Chemicals, Osaka, Japan), thoroughly mixing them in a crucible for calcination for 4 h in a muffle furnace, and then cooling them to room temperature.

### 3.2. Transesterification

Into a cone flask with a cooled condenser, 12.5 g (0.0114 mol) of vegetable oil (Great Wall Business Co., Taiwan) was added, with various methanol (American Chemical Society grade, ECHO Chemical Co., Miaoli, Taiwan) to oil molar ratios (12-24) and catalyst quantities (2-10 wt.%) at 338 K for 4 h of transesterification. All of the experiments were performed under atmospheric pressure. An appropriate amount of deionized water was added into the sample to stop the transesterification reaction. After a period of time, the sample was cooled and separated through a typical filter paper. The extra methanol was removed from filtrate and water prior to the FAME analysis. The FAME composition of the biodiesel was determined by gas chromatography.

### 3.3. Characterization

The alkali strength analysis of the bauxite used as a precursor to prepare catalyst (H⁻) was defined as the acid–base indicator. All alkali catalysts were characterized using an X-ray diffraction instrument with Cu K$\alpha$ radiation (MAC MXP18, Tokyo, Japan). The structure of the as-prepared alkali was observed employing a field emission scanning electron microscope (FE-SEM, JEOL JSM-6700F). FAME composition in the biodiesel was determined by using a Thermo trace gas chromatograph with a flame ionization detector. A Tr–biodiesel (30 m × 0.25 mm × 0.25 μm film thickness) capillary column was used. Samples were injected under the following conditions: the carrier gas was Nitrogen with a flow rate 2 mL/min., an injector temperature was 200 °C was 90 °C for 0.5 min and increased to 260 °C (programmed temperature) at a rate of 10 °C/min, and the detector temperature was 250 °C. This procedure was developed according to EN 14103.

## 4. Conclusions

Bauxite was found to contain Si and Al essential compounds suitable as catalysts for biodiesel production. A highly effective catalyst for biodiesel production was synthesized using bauxite and $Li_2CO_3$ and by calcination at temperatures of 800 °C for 4h. The transesterification procedure of biodiesel reached over 96% and the optimal parameters were 65 °C reaction temperature, 3 h reaction time, 9 : 1 methanol-to-oil molar ratio and 3 wt.% catalyst amount. Under the optimal parameters, the catalyst can be effectively reused for 6 runs with a minimal decrease of <10% in the conversion rate. In addition, the results demonstrate that the efficiency with high free fatty acid concentration was significantly influenced. Consequently, the efficiency of the same catalyst under the same reaction conditions differs with different oil. In this work, the application of bauxite as a catalyst for biodiesel production not only provides a cost-effective and environmentally friendly way of using the bauxite, but also reduces the cost of biodiesel production.

**Author Contributions:** Data conceptualization, Investigation, Writing-original draft, Methodology, Yong-Ming Dai. Methodology, Validation, Investigation, Software, Cheng-Hsuan Hsieh and Jia-Hao Lin. Writing-original draft preparation, Fu-Hsuan Chen. Supervision, Conceptualization, Visualization, Writing – review, Editing, Validation, Chiing-Chang Chen.

**Funding:** This research was supported by the Ministry of Science and Technology of the Republic of China (MOST-108-2113-M-142-001).

**Acknowledgments:** We thanks for the financial support of this research by Ministry of Science and Technology of Taiwan. We also thank for the Center of Expensive Instruments at National Chung Hsing University to perform FE-SEM analysis.

**Conflicts of Interest**: The authors declare that they have no known competing financial interests or personal relationships that could have appeared to influence the work reported in this paper

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
