# Peer review of "Biodiesel Production Using Bauxite in Low-Cost Solid Base Catalyst Precursors"

_catalysts, doi:10.3390/catal9121064_

Round 1

Reviewer 1 Report

Dear authors,

Please adjust figure 2 because is distorted.

Please revise the introduction section.

Best regards!

Author Response

Reviewer 1 comments:

Please adjust figure 2 because is distorted.

Answer: As the reviewer suggested, Figure 2 has been revised.

Please revise the introduction section.

Answer: As the reviewer suggested, the introduction section has been revised.

Reviewer 2 Report

The article has been slightly improved, however, the quality of the results, the information it contains and the conclusions obtained are the same. The response of the author to my comments have not been satisfied, without giving any reason or explanation. I maintain my opinion that the authors should send the article to a journal with a lower impact factor.

Author Response

Reviewer 2 comments:

The article has been slightly improved, however, the quality of the results, the information it contains and the conclusions obtained are the same. The response of the author to my comments have not been satisfied, without giving any reason or explanation. I maintain my opinion that the authors should send the article to a journal with a lower impact factor.

We would like to thank the reviewer for careful and thorough reading of this manuscript and for the thoughtful comments and constructive suggestions, which help to improve the quality of this manuscript.

As suggested by the reviewer, we have reviewed carefully the entire manuscript and have added the contents of this paper , as shown in the revised manuscript.

Reviewer 3 Report

I think that the subject of the article is interesting, as it is always desirable to find alternatives to reduce the use of conventional catalysts in biodiesel production in the most profitable way.

Having said that, I find some drawbacks to this research work, as the following, that should be considered before the possible final acceptance of the paper:

First of all, the English is not (yet) good enough for publication in your journal. I highly recommend the help of a native speaker for further revision of the whole text. This way, your work will be more understandable. I think that the literature of the article is “old” and not enough to justify the results. Add more modern literature about the subject, especially in the introduction and result and discussion sections, to support the justification of your work and your final results. The procedure of biodiesel production, although typical, should be more detailed. For instance, did you put every reactive at once? This kind of detail could be useful to "reproduce" your experiment by other colleagues.  I also miss some results about the biodiesel obtained. I understand that you are focused on the yield of the process, but some quality parameters could be useful (at least for the “final biodiesel” obtained with the best conditions. Only by paying attention to the yield of FAMEs, the comparison with standards would be perfect to discuss these results, and according to them, give an approach of the possible use of the biodiesel obtained according to standards such as UNE-EN 14214 or equivalent. You need to compare the reusability of your catalyst with others (I saw in the literature that the reusability obtained is average, so this would be useful to support your work). In the conclusion section, you state that the cost of biodiesel production is reduced by using this catalyst. What comparison did you do to conclude that? With other conventional basic or acidic catalysts? To do so, you need to assess the amount of bauxite and other catalyst, temperature conditions, reaction times, and so on. So go further in this point or delete this statement, as you cannot prove it in the current work (as far as I can tell).

Author Response

Answer: As the reviewer suggested, the manuscript  has been revised.

We have revised the manuscript carefully and tried to avoid any grammar or syntax error. In addition, we have asked several colleagues who are skilled authors of English language papers to check the English.

Round 2

Reviewer 2 Report

After I read carefully and thoroughly this manuscript, my thoughtful comments and constructive suggestions are the authors should publish the research in another journal with less impact factor. This piece of researcher has not the quality of this journal.

This manuscript is a resubmission of an earlier submission. The following is a list of the peer review reports and author responses from that submission.

Round 1

Reviewer 1 Report

Authors proposed a new and simple method of Biodiesel production using bauxite and Li2CO3 as material to prepare the solid base catalyst.
The manuscript must be correctly rewritten and / or reorganized

For example:

(1) Figure 1 and 2 show XRD images but the caption is not correct (please to change the caption respectively);
(2) In the Reaction studies
- lines 123 page reads “Figure 4 shows different calcination temperature and time of bauxite and Li2CO3 mole ratio…”, Figure 4 is not correct
please change Figure 4 to Figure 1
In the same way:
- lines 126 reads “...is higher than 800 °C, as shown in Figure 4", Figure 4 is not correct, please change Figure 4 to Figure 2

(3)…etc

Reviewer 2 Report

Dear authors,

Please follow the next observation:

Abstract section:

The abstract section is the first section (besides the title) that a reader consults. Thus in my opinion the research conducted in this manuscript should be summarized more adequate. In addition, please underline better the aim of this paper and also the state of the art; Some numerical conclusions would be appreciated.

Introduction section:

It seems that the topic presented in this manuscript is not research enough. Please expand the stat of the art research; Which are in fact the disadvantages and the advantages of the proposed method? Some information are given but are not enough underlined; Please underline the novelty of this research.

Experimental methods and chemical materials section:

Why the need to give the manufacturing information? Please explain better the method;

Result section:

Please explain better the results. More precisely the XRD test. Please use only English words. Also, high resolution figures is needed (figures 1 and 2); The titles for figures 1 and 2 are switched? Please be consistent. Choose one acronym: FESEM or FE-SEM; Page 5, line 114: “According to the figure”. What figure? Figure 3 has a low resolution. Please resolve this issue. Page 5, line 117: “on the conversion eaction conditions”. I supposed that you meant reaction? Pages 4 and 5, lines 111 and 112: “The catalyst has the uniform micro scale block and particle mixed composition with 20-50 μm in size as shown in Figure 3”. Please highlight this information on figure 3. Page 5, lines 123 and 124: “Figure 4 shows different calcination temperature and time of bauxite and Li2CO3 mole ratio is 4 123 through the transesterification process”. Where in figure 4 are presented different calcination temperatures? Page 5, line 136: “Consequently, it was required to select a correct calcination tim to promise completion”. The correct word is time? Page 6, line 153 and 154. The correct form for weight percent is wt.%.

Conclusion section:

Please expand the conclusion.

Best regards!

Reviewer 3 Report

The article looks like a short communication. 6 of the 8 pages that are composed, it is title, introduction, materials, methods, tables and figures. The latter of poor quality, with letters in Chinese, with imperfections and very large. The characterization of solids is very poor, as is their discussion. The catalytic reactivity of the reactions is around 95%, it does not matter the catalyst, the conditions and the reaction time and the authors draw conclusions about these data that are not correct.
My opinion is that the authors send the work to a magazine with a lower impact index.